# iDNA-ITLM: An interpretable and transferable learning model for identifying DNA methylation

**Xia Yu[1,2], Cui Yani[1], Zhichao Wang[3], Haixia Long[2], Rao Zeng[2], Xiling Liu[2], Bilal Anas[2], Jia Ren [1]***

**1** School of Information and Communication Engineering, Hainan University, Haikou, Hainan, China, **2** Key Laboratory of Data Science and Smart Education, Ministry of Education, Hainan Normal University, Haikou, Hainan, China, **3** Unit 32033, The People's Liberation Army, Beijing, China

* 7403491@qq.com

**Data Availability Statement:** Data and code are available publicly and can be found in the Supporting information files uploaded alongside this manuscript. Data and code have also been

## Abstract

In this study, from the perspective of image processing, we propose the iDNA-ITLM model, using a novel data enhance strategy by continuously self-replicating a short DNA sequence into a longer DNA sequence and then embedding it into a high-dimensional matrix to enlarge the receptive field, for identifying DNA methylation sites. Our model consistently outperforms the current state-of-the-art sequence-based DNA methylation site recognition methods when evaluated on 17 benchmark datasets that cover multiple species and include three DNA methylation modifications (4mC, 5hmC, and 6mA). The experimental results demonstrate the robustness and superior performance of our model across these datasets. In addition, our model can transfer learning to RNA methylation sequences and produce good results without modifying the hyperparameters in the model. The proposed iDNA-ITLM model can be considered a universal predictor across DNA and RNA methylation species.

## 1. Introduction

DNA methylation is crucial in numerous biological processes and has been linked to several diseases, notably cancer [1,2]. Recognizing DNA methylation sites accurately is significant in understanding gene regulation and disease mechanisms. Deep learning methods have gained prominence in DNA methylation site recognition in recent years, offering promising results. Currently, three extensively researched DNA methylation species are recognized: N4-Methylcytosine (4mC), 5-hydroxymethylcytosine (5hmC), and N6-methyladenine (6mA) [3].

In 6mA prediction, the algorithm sNNRice6mA utilized a two-dimensional one-hot encoding technique for DNA sequences. It utilized a convolutional neural network (CNN) model to make predictions about 6mA sites [4]. Tsukiyama et al. harnessed Word to Vector (word2vec) and Bidirectional Encoder Representations from Transformers (BERT) technologies, constructing the deep learning-oriented framework BERT6mA, which delivered a remarkable results in forecasting 6mA modification sites [5]. Rehman et al. presented a framework based

uploaded to GitHub at the following URLs: Data available link: https://github.com/Yyxx-1987/iDNA-ITLM/tree/master/iDNA-ITLM/data Code download link: https://github.com/Yyxx-1987/iDNA-ITLM/tree/master/iDNA-ITLM All data and code can also be obtained from the author via email at 408423952@qq.com.

**Funding:** This work is supported by the National Natural Science Foundation of China (No.62302132, No.62262016, No.61961160706, No.62262018, No.62262019), 14th Five-Year Plan Civil Aerospace Technology Preliminary Research Project(D040405), the Hainan Provincial Natural Science Foundation of China (No.823RC488, No.623RC481, No.620RC603, No.621MS038), the Haikou Science and Technology Plan Project of China (No.2022-016), the Program of Hainan Association for Science and Technology Plans to Youth R & D Innovation (QCQTXM202209), the Project supported by the Education Department of Hainan Province (Hnky2024-18).

**Competing interests:** The authors declare no competing interests.

on CapsuleNet for identifying DNA m6A sites, demonstrating its effectiveness in accurately predicting these methylation sites [6]. Tsukiyama et al. demonstrated using BERT-based models to enhance the precision of 6mA site prediction in DNA and its efficacy in accurately identifying these methylation sites within DNA sequences [7]. Hasan et al. presented Meta-i6mA, an integrative machine-learning framework designed by harnessing informative features for cross-species prediction of DNA 6mA sites, specifically in the genomes of plants. It effectively addresses interspecies variations, providing a versatile tool for plant genome research and epigenetic analysis [8]. Chen et al. introduced an innovative approach, DeepM6ASeq-EL, that combines Long Short Term Memory and ensemble learning techniques to accurately predict human m6A sites in RNA methylation sites. Fusing these methods enhances the model's predictive performance, providing a robust tool for identifying m6A modification sites in the human genome [9]. Zhang et al. integrated the attention mechanism into the model, helping capture and prioritize important features, leading to more effective detection of these epigenetic modifications in DNA sequences [10].

In 5mC prediction, A distinct approach, named DIRECTION, amalgamated Feature selection guided by beam search. with traditional machine learning algorithms of anticipating 5mC residues [11]. Tran TA et al. Employed a technique for feature extraction, utilizing k-mers embeddings derived from a language model that has undergone training [12]. BiLSTM-5mC employed both the one-hot encoding and the nucleotide property and frequency (NPF) methodologies to represent nucleotide sequences. Subsequently, it incorporated the bidirectional long short-term memory (BiLSTM) model and a fully connected network to predict the methylation sites [13].

In 4mC prediction, considerable attention has been directed. In 2019, two algorithms, 4mCCNN [14] and 4mCPred-SVM [15], emerged for forecasting 4mC residues. 4mCCNN was rooted in CNN, while 4mCPred-SVM was built upon SVM (support vector machine). Furthermore, Liu et al. proposed DeepTorrent, an combined model combining CNN and BiLSTM and to detect 4mC sites [16]. Another algorithm, Deep4mC, demonstrated the efficacy of a standalone CNN model in achieving commendable 4mC prediction results [17]. Additionally, Hyb4mC proposed that an elastic net coupled with a capsule network proved effective for small datasets, whereas CNN excelled with larger ones [18]. Zeng et al. introduced a two-layer deep learning architecture named Deep4mcPred, incorporating a blended network composed of ResNet and Long Short-Term Memory (LSTM) [19].

Nonetheless, most current methodologies can solely differentiate a singular form of DNA methylation, rendering them challenging to extend to other types of methylation [20]. However, only a limited number of approaches tackle all three categories of methylation as described earlier [21–25], notably iDNA-MS [39], iDNA-ABT [37], and iDNA-ABF [38].

DNA methylation datasets suitable for deep learning typically consist of relatively short sequence lengths per sample, with 41 base pairs (bp) being the most common. In deep learning models, DNA methylation sequences are often utilized to extract semantic information, but due to their short length, the extracted semantic information may be insufficiently rich. Extracting more comprehensive semantic sequence information and enhancing the classification accuracy of methylation sites within limited data volume is also a prominent challenge in the current field of DNA methylation site identification.

The primary objective of this article is to present iDNA-ITLM to learn biological contextual semantics. This innovative deep-learning framework integrates five modules to collectively predict methylation sites for three distinct modification types. In contrast to the existing approaches, iDNA-ITLM brings forth the following contributions.

1. Our model introduces an interpretable new data enhancement strategy; each DNA methylation sequence is self-replicated multiple times and then embedded into a high-dimensional

matrix, much like a image, in order to extract features. The size of the transformed matrix is magnitudes greater, amounting to several thousand times the size of the original sequence sample.

2. Our model has the capability to predict 17 distinct DNA methylation species encompassing three types of DNA methylation modifications (4mC, 5hmC, and 6mA), and the model of iDNA-ITLM can achieve good predictive performance.

3. Our model can transfer learning to RNA methylation sequences and produce good predictive results without modifying the hyperparameters in the model.

4. Our model can be considered a universal predictor employing identical hyperparameters across all DNA and RNA methylation species.

## 2. Methods

### 2.1 Overview of iDNA-ITLM

Fig 1 presents comprehensive overview of our model iDNA-ITLM. The process of collecting the dataset is represented in Fig 1A. The workflow of iDNA-ITLM is delineated in Fig 1B–1F,

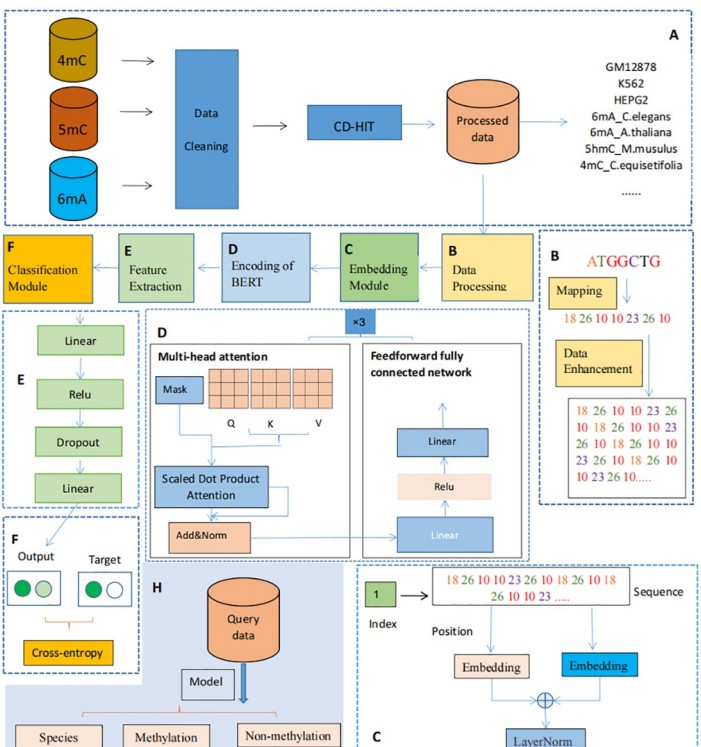

**Fig 1. The overview of iDNA-ITLM.** The A module depicts the reorganization of DNA methylation datasets into training and independent datasets, categorized based on the three primary DNA methylation types. The Modules of B to F provide a comprehensive representation of the iDNA-ITLM architecture. B corresponds to the data processing module, which employs efficient mapping and data enhancement techniques to process input sequences. C corresponds to the Embedding Module, which dynamically acquires relevant embeddings. D corresponds to the BERT Encode Module, which utilizes BERT encoders to encode DNA sequences. E corresponds to the Feature Extraction Module, which is responsible for extracting the final features. Lastly, F corresponds to the Classification Module, which incorporates cross-entropy for predictive tasks. H outlines the workflow for interpretable analysis during testing. In our model, we utilize attention mechanisms to capture and understand sequential patterns within query sequences.

encompassing five principal modules: (B) data processing, (C) Embedding, (D) Encoding, (E) Feature extraction, and (F) Classification. The prediction process is delineated in the following manner: In module B (refer to Fig 1B), we utilize DNA sequences to self-replicate multiple times and a mapping-based tokenization scheme to process DNA sequences. Subsequently, in module C (as shown in Fig 1C), the iDNA-ITLM employs an Adaptive Embedding approach to associate each of the four nucleotide characters with a vector, transforming the tokenized samples into a two-dimensional matrix. From the image processing perspective, the purpose of doing this is to continuously self-replicate a short DNA sequence into a longer DNA sequence and then embed it into a high-dimensional matrix to enlarge the receptive field to facilitate the model's learning of rich features. This is like the receptive fields of a small-scale image and a large-scale image being different, and the learned features are also different.

Module D leverages a combination of Masked Language Model (MML), Multi-head Attention and a Feedforward Fully Connected Network of BERT to produce the evolutionary output feature. Following this, within module E (depicted in Fig 1E), a fully connected network comprising four layers, including two fully connected layers, a dropout layer, and a ReLU layer, is utilized to extract features from methylated and non-methylated DNA sequences, effectively mitigating overfitting concerns. In module F, we employ the Adam optimizer [37] for training due to its efficiency, memory-friendliness, and suitability for models with numerous parameters. Cross-entropy [38] is used to quantify the disparity between predicted and target outcomes. This methodology also involves tuning the model's parameters to minimize the discrepancy between predicted probabilities and actual labels. It's worth emphasizing that we elaborate on the specifics of these four modules in the subsequent sections.

## 2.2 Data processing module

DNA sequences consist of textual strings using the bases (A, C, T, G). Typically, these sequences are transformed into numerical attributes to suit machine learning methodologies. In deep learning models, DNA methylation sequences are often utilized to extract semantic information, but due to their short length, the extracted semantic information may be insufficiently rich.

**Sequences representation.** We represent a DNA sequence through tokenization with the representations of a mapping (dictionary) between amino acid residues (or nucleotide residues in DNA/RNA sequences) and their corresponding numerical indices. Such a mapping is often used to convert sequences of biological molecules (like proteins or DNA) into numerical representations that machine learning algorithms or neural networks can process.

$$
\begin{aligned}
&\text{Token2index} = \{ \\
&\text{CLS}, 1 \\
&\text{SEP}, 2 \\
&\text{MASK}, 3 \\
&\cdots \\
&\text{A}, 18 \\
&\text{G}, 10 \\
&\text{C}, 23 \\
&\text{T}, 26 \\
&\cdots \\
&\}
\end{aligned}
\tag{1}
$$

As an illustration, consider the DNA sequence "ATGGCTG", which can be tokenized as a sequence of tokens, such as "18 26 10 10 23 26 10".

**Data enhancement.** We process each tokenized sample (DNA tokenized sample sequence of length 41) as follows. Using self-replication multiple times of a tokenized sample, each tokenized sample is represented by $Sublist_i$, the multiple self-replications of the tokenized sample represented by $Mul\_replicate_i$, the calculation expression is as follows: n represents the number of repetitions, mul_replicate represents the sample sequence after data enhancement, and i represents the i-th sample.

$$\mathrm{MuLreplicate}_i = (\mathrm{Sublist}_i)_1 + (\mathrm{Sublist}_i)_2 + ... + (\mathrm{Sublist}_i)_n \tag{2}$$

## 2.3 Embedding module

In the embedding module, we adopt the Adaptive embedding method [26], which associates every one of the four nucleotide characters with a vector. This is achieved by adding a dedicated randomly initialized vector retrieved from the lookup table to the vector's position within the sequence. Subsequently, each vector dynamically refines its values based on the given task through backpropagation while the model undergoes training. The multiple replicates tokenized sample is embedded in a two-dimensional matrix; the size of the embedded matrix is $d_l \times d_m$. $d_l$ represents the length of multiple self-replication tokenized sample, $d_m$ represents the dimension of each sample after embedding, The value of $d_m$ is large enough; this is like the receptive fields of a small-scale image and a large-scale image being different, and the learned features are also different. The explanation for this embedding process is detailed below:

$$\mathrm{Embedding} = \mathrm{embed}_{\mathrm{token}}(\mathrm{MuLreplicate}_i) + \mathrm{embed}_{\mathrm{pos}}(\mathrm{MuLreplicate}_i) \tag{3}$$

$$\mathrm{Embedding\_data} = \mathrm{Layer\_norm}(\mathrm{Embedding}) \tag{4}$$

$embed_{pos}$ signifies the embedding of the DNA sequence's position. $embed_{token}$ corresponds to the embedding for the DNA sequence. $Mul\_replicate_i$ symbolizes the *i-th* DNA sequence after data enhancement. Layer_norm stands for Layer normalization, a technique that normalizes the feature dimensions of each sample. This helps decrease internal covariate shifts and expedites the training of neural networks.

## 2.4 Encoding process of BERT

BERT constitutes a bidirectional language representation model constructed based on the transformer architecture first introduced in [27] and has found extensive applications in numerous NLP tasks. In this context, we employ a pre-trained BERT model, DNABERT [28], consisting of three transformer layers, each with 768 hidden units and featuring eight attention heads per layer. Given the absence of inherent semantic logic within DNA sequences, we adopt increasing the length of DNA methylation sequence and enhancing model through data enhancement and embedding into large-scale matrices. Within the module, we employ a masked language modeling approach similar to the one employed in the original BERT.

**Masked language Model(MLM).** In the MLM task, some words or subwords in the text sequence are randomly selected and replaced with a special "[MASK]" token. The model aims to predict the substituted word within the context. The model is compelled to predict the correct word in the missing context through this pretraining task, thus learning bidirectional contextual information. By using masks during pretraining, BERT can simultaneously consider

each word's left and right context information, thus better capturing the dependencies between words. Traditional language models with autoregressive properties, like recurrent neural networks and the decoder part of Transformers typically only use left or right context. However, with the mask task, BERT enforces the model to predict the missing context during training, thereby alleviating bias issues in autoregressive models. By masking certain words during pretraining, BERT can access a variety of language patterns, thus better-capturing language diversity and complexity, which enhances its generalization capability across various downstream tasks.

We create a masking matrix where the values are set to 1 for padding token positions while the values at other positions remain 0. During attention computation, perform element-wise multiplication with the attention weight matrix. This leads to attention weights of 0 at padding token positions, excluding them from consideration and ensuring that the model doesn't allocate attention to padding tokens. Padding tokens do not affect the computation of attention weights for other genuine tokens. This contributes to enhancing the efficiency and performance of the model.

**Attention calculation.** We calculate the scalar product of the query vector (Q) and the key vector (K), then scale the outcome to prevent the attention scores from becoming excessively large while also incorporating a scaling factor (typically the reciprocal square root of the dimension of the key vectors). The attention scores undergo a softmax operation to attain normalized attention weights. These attention weights are employed for a weighted summation across the value vectors (V), culminating in the ultimate attention representation. The self-attention's mathematical depiction is outlined as follows:

$$\begin{cases} Q = XW^Q \\ K = XW^K \\ V = XW^V \end{cases} \tag{5}$$

$$\mathrm{Self-attention}(Q, K, V) = \mathrm{softmax}\left(\frac{QK^T}{\sqrt{d_k}}\right)V \tag{6}$$

Here, $X \in R^{L*d_m}$ represents the resultant embedding derived from the module of embedding, where $d_m$ Signifies the dimension of embedding, and L is input sequence's length. Q, K, V $\in R^{L \times d_k}$ Denote the query, key and value matrix, respectively. These matrices are acquired from X via a linear transformation involving. $W^Q$, $W^K$, and $W^V$, all of which belong to the real space $R^{d_m*d_k}$. In this context, $d_k$ Signifies the dimension of query, key, and value vectors. $d_m$ and $d_k$ both are considered hyperparameters.

**Multi-head attention.** In this context, the computation of the attention head indexed by 'i' is performed in the subsequent manner:

$$Q_i = XW_i^Q, K_i = XW_i^K, V_i = XW_i^V, i = 1, \cdots, h \tag{7}$$

$$Head_i = Self-attention(Q_i, K_i, V_i) \tag{8}$$

$$\mathrm{MultiHead-Attention}(Q, K, V) = \mathrm{Concact}(Head_1, Head_2, \cdots, Head_h)W^O \tag{9}$$

$W_i^Q$, $W_i^K$, and $W_i^V \in R^{d_m \times d_k}$ Stand for the query, key and value matrices of the *i-th* head. The parameter 'h' represents the quantity of heads. Subsequently, we employ multi-head attention on Q, K and V by concatenating 'h' heads, each utilizing self-attention with respect to the

input sequence. Additionally, $W^o \in R^{d_m \times d_k}$ Functions as a linear transformation matrix, mapping the multi-head attention's resulting dimensions to match the encoder block's input dimensions. This facilitates a skip connection, connecting the input to the encoder block interacts with the output of the multi-head attention mechanism.

**Feedforward fully connected network.** We perform operations on the output of every multi-head attention block. The process involves projecting the attention representations from each position within the input sequence (after merging self-attention computations and multi-head attention) into a higher-dimensional feature space. Subsequently, an activation function like ReLU is used to introduce nonlinearity, and then the data is projected back to its original dimension. This enhances the model's capacity to capture features effectively.

$$\text{Linear\_output} = \text{W\_1X} + \text{b\_1} \tag{10}$$

$$\text{Relu\_output} = \text{Relu(Linear\_output)} \tag{11}$$

$$\text{ffn\_output} = \text{W\_2} * \text{Relu\_output} + \text{b\_2} \tag{12}$$

Here, W_1 and W_2 stand for matrices of weights, while b_1 and b_2 denote bias vectors, X represents the resultant derived from the multi-head attention module. Activation refers to the activation function, and the commonly used activation function in feedforward neural networks is the Rectified Linear Unit (ReLU) function. Each encoder layer has an independent feedforward neural network.

## 2.5 Feature extraction module

Subsequently, the extracted feature vectors undergo input into a fully connected network comprising four layers. The initial and final layers of this network are both linear transformation models. The second layer incorporates a dropout mechanism, which aids in counteracting overfitting, while the third layer employs a rectified linear unit (ReLU) activation. This ensemble of four layers collaborates in predicting two classes: samples exhibiting methylation and those without methylation.

## 2.6 Classification module

The model used the Adam optimizer [29] for training, which is known for its efficiency, requires fewer memory resources and is well-suited for handing model with large parameters. We employ cross-entropy [30] for the binary classification task to quantify the difference between the predicted and target outcomes. This approach also involves tuning the model's parameters to minimize the disparity between predicted probabilities and actual labels. This optimization enhances the model's predictive accuracy for DNA methylation classification. The function is provided below:

$$p_k = \frac{exp\left(y_{p,k}\right)}{\sum_j exp\left(y_{p,j}\right)}, k = 0, 1 \tag{13}$$

$$L_{CE}(p_1, y) = -y \log p_1 - (1 - y)\log (1 - p_1) \tag{14}$$

Here, k can take the values 1 or 0, representing either the DNA methylation or non-methylation classes. In this context, y stands for the true label, while $p_k$ signifies the probability assigned to the sequence being categorized as class k.

## 3. Performance metrics

We assess how well iDNA-ITLM model and other pre-existing models perform using the subsequent five frequently employed measurements [31–34]: Accuracy (ACC), Sensitivity (SN), Specificity (SP), Matthews' correlation coefficient (MCC) and Area Under Curve (AUC). The equations for these measurements are expounded below:

$$ACC = \frac{TP + TN}{TP + FN + TN + FP} \tag{15}$$

$$SN = \frac{TP}{TP + FN} \tag{16}$$

$$SP = \frac{TN}{TN + FP} \tag{17}$$

$$MCC = \frac{TP \times TN - FP \times FN}{\sqrt{(TP + FN)(TP + FP)(TN + FP)(TN + FN)}} \tag{18}$$

$$AUC = \frac{\sum_{i \in pos} rank_i - \frac{num_{pos}(num_{pos}+1)}{2}}{num_{pos} num_{neg}} \tag{19}$$

Here, TP, FN, TN and FP denote the counts of true positive, false negative, true negative and false positive instances, respectively. ACC and MCC both are employed for gauging the model's comprehensive performance. SN pertains to the ratio of accurately predicted samples correctly identified as methylated by a predictive model, while SP quantifies the proportion of accurately predicted non-methylated samples by the model. The Area Under Curve (AUC) is determined as the region enclosed between the Receiver Operating Characteristic Curve (ROC) and the coordinate plane, where the false positive rate (FPR) is plotted on the x-axis and the true positive rate (TPR) is plotted on the y-axis. In total, an increase in these metrics signifies an improved model performance.

## 4. Results

The model iDNA-ITLM is developed utilizing Keras version 2.9.0 and TensorFlow version 1.12.0 with Python 3.9. The model undergoes training through a 5-fold cross-validation process, with each fold trained over the course of 100 epochs, employing a batch size set 64, learning rate set 0.0001, an embedding size of 448, and a dropout of 0.4.

### 4.1. Benchmark datasets

We opt for the datasets from the iDNA-MS web server [35] encompassing training and independent testing subsets to assess the accuracy and suitability of our model, as illustrated in Table 1. Our experimental evaluations in this study involve 17 datasets comprising 501,200 DNA sequences. Specifically, the DNA sequences associated with the 6mA, 5hmC and 4mC modifications are sourced from iDNA-MS, and all of these collected sequences are 41 base pairs in length. Notably, among the 6mA samples, methylated adenine (A) consistently occupies the central position among the 5hmC and 4mC samples, while methylated cytosine (C) occupies the central position. Negative samples, on the other hand, adhere to specific criteria: they are unreported for methylation and contain Cytosine or Adenine at the central positions.

**Table 1. Overview of the datasets.**

| ID | Dataset | Training | | Independent Testing | |
|---|---|---|---|---|---|
| | | Positive | Negative | Positive | Negative |
| 1 | 4mC_C.equisetifolia | 183 | 183 | 183 | 183 |
| 2 | 4mC_F.vesca | 7899 | 7899 | 7898 | 7898 |
| 3 | 4mC_S.cerevisiae | 990 | 990 | 989 | 989 |
| 4 | 4mC_Tolypocladium | 7664 | 7664 | 7663 | 7663 |
| 5 | 5hmC_H.sapiens | 1172 | 1172 | 1172 | 1172 |
| 6 | 5hmC_M.musculus | 1840 | 1840 | 1839 | 1839 |
| 7 | 6mA_A.thaliana | 15937 | 15937 | 15936 | 15936 |
| 8 | 6mA_C.elegans | 3981 | 3981 | 3980 | 3980 |
| 9 | 6mA_C.equisetifclia | 3033 | 3033 | 3033 | 3033 |
| 10 | 6mA_D.melanogaster | 5596 | 5596 | 5595 | 5595 |
| 11 | 6mA_F.vesca | 1551 | 1551 | 1551 | 1551 |
| 12 | 6mA_H.sapiens | 9168 | 9168 | 9167 | 9167 |
| 13 | 6mA_R.chinensis | 300 | 300 | 300 | 300 |
| 14 | 6mA_S.cerevisiae | 1893 | 1893 | 1893 | 1893 |
| 15 | 6mA_T.thermophile | 53800 | 53800 | 53800 | 53800 |
| 16 | 6mA_Tolypocladium | 1690 | 1690 | 1689 | 1689 |
| 17 | 6mA_Xoc BLS256 | 8608 | 8608 | 8607 | 8607 |

In Table 1, the "Dataset" column represents the name of each dataset. In the dataset names, the separator "-" before indicates the type of methylation modification, while the name after the separator represents the species type. Details regarding the quantities of positive and negative samples can be found in the "Training," and "Testing" columns, respectively.

We performed a DNA methylation analysis encompassing three distinct methylation Modification types: 4mC, 5hmC and 6mA. Utilizing the WebLogo tool [36], each of the three types of DNA methylation modifications is selected, and we plot the distribution of DNA methylation sequences, as depicted in Fig 2. Upon comparing these Logo plots, we note significant distinctions in the positions and extents of base enrichment within the genome among the diverse methylation types. For instance, in the species H. sapiens with 5hmC, the central base is CG, whereas in the species C. equisetifolia with 4mC, the central base is C, and in the species A. thaliana with 6mA, the central base is A. In the species H. sapiens with 5hmC, there is a higher enrichment of C, G bases, while in the same species, there is a higher enrichment of A, T bases. Conversely, in the species A.thaliana with 6mA, the enrichment levels of ATCG bases are relatively balanced.

Within each Logo plot for every DNA methylation type, discernible conservative and variable patterns are evident in specific regions. These patterns potentially mirror their biological functions and regulatory mechanisms across different species. These findings highlight the complexity of various DNA methylation types and their potential implications in gene expression and epigenetic regulation.

## 4.2 data enhancement

**4.2.1 The visualization of data enhancement.**   The enhanced DNA methylation data is marked in Fig 3, and Fig 3 shows the data enhancement visualization maps of 4mc_F.vesca species, 5hmC_H.sapien species and 6mA_H.sapiens species are depicted separately. In the figure, the regions highlighted by the red and yellow boxes are the embedding areas generated

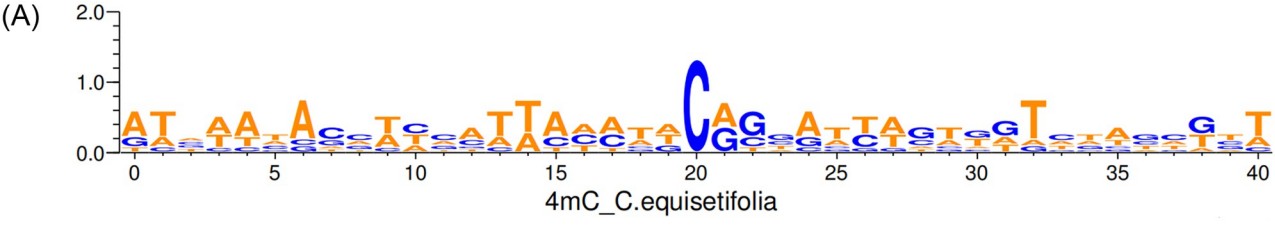

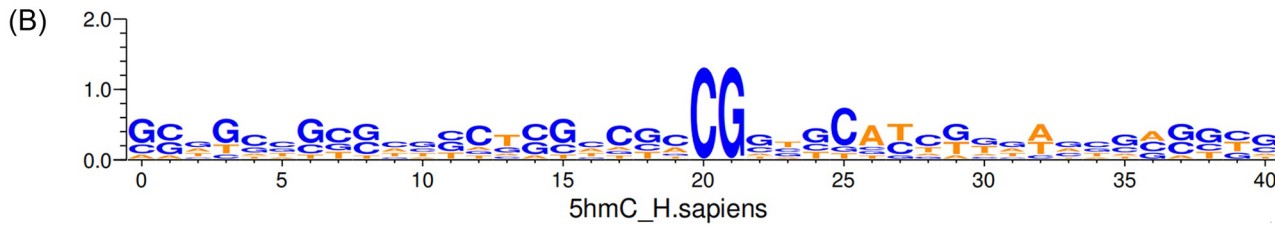

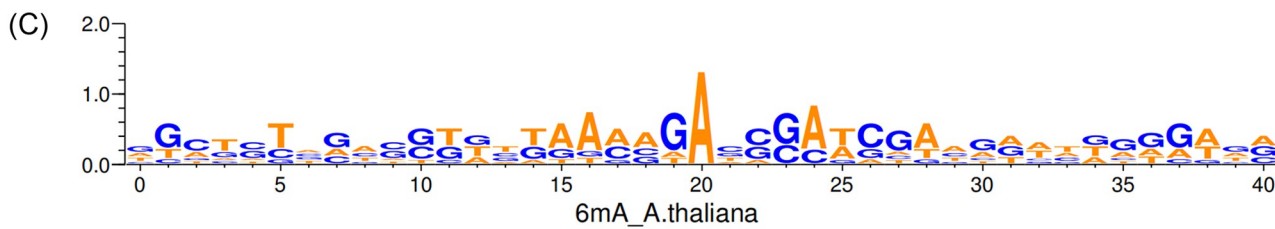

**Fig 2. The Logo of three distinct types of methylation.**

after the self-replication of the DNA sequences. The visualized shapes of the data after embedding are identical. This significantly enhances the machine's perceptiveness.

Using Self-replication of DNA methylation sequence data enhancement can increase the amount of training data, which is particularly useful in cases of limited data availability, as it can enhance the model's attention to features. This is similar to how humans when recognizing an object, may not have a strong memory with just one glance. Still, if they see the same object multiple times, it strengthens the brain's memory and teaches more about the object's features. Enhancing DNA methylation sequence data works similarly, improving the model's performance and generalization capability.

**4.2.2 The model performance with and without data enhance strategy.** We use the ROC curve to assess the capability of our model, with the ROC curves before and after data augmentation shown in Fig 4. The ROC (Receiver Operating Characteristic) curve is a crucial tool for measuring the performance of classification models. It evaluates the model's ability to distinguish between classes by comparing the true positive rate (TPR) and the false positive rate (FPR). A higher area under the ROC curve (AUC) indicates better classification performance of the model.

For testing, we randomly selected a species from each of the three different categories (4mC, 5hmC, 6mA), the randomly selected species are 4mC_F.vesca, 5hmC_H.sapiens, and 6mA_H.sapiens species., and Fig 4 displays the ROC curves for these species. It is evident from the figure that the ROC curve values significantly change after applying data augmentation. Compared to before the application of data augmentation strategies, the area under the ROC curve is larger with data augmentation. For example, for the 4mC_F.vesca, the ROC value was

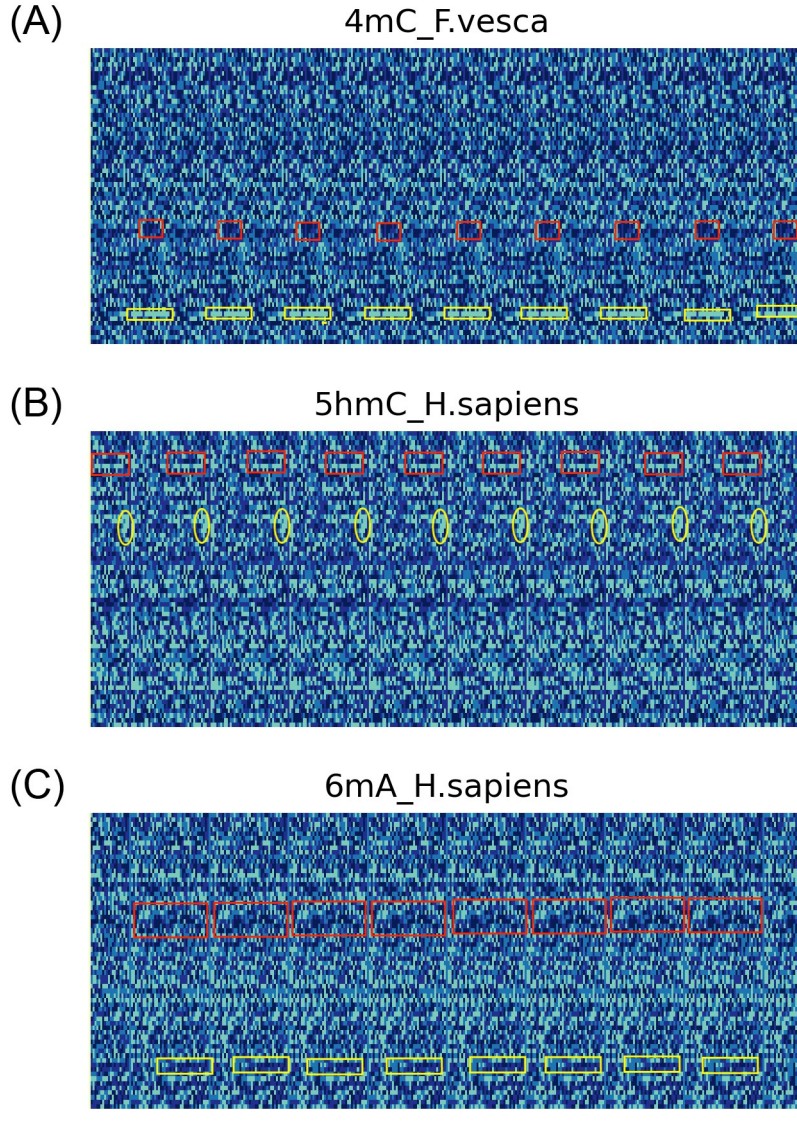

**Fig 3. The data enhancement visualization of DNA sequence.**

0.89 before using data augmentation strategies, and it increased to 0.92 after the application of data augmentation. This indicates that data augmentation strategies can, to a certain extent, enhance the performance of the model.

### 4.3 Visual feature representation through dimensionality reduction in the course of training

To visually illustrate the efficacy of our model, we employ Uniform Manifold Approximation and Projection (UMAP) [37] to condense the feature space into a two-dimensional representation of the DNA methylation datasets, which are partly illustrated in Fig 5. UMAP is a non-linear dimensionality reduction technique crafted to convert high-dimensional data into a lower-dimensional space while preserving both local and global relationships among data points. As illustrated in Fig 5, it becomes evident that during epoch 1, positive and negative samples are

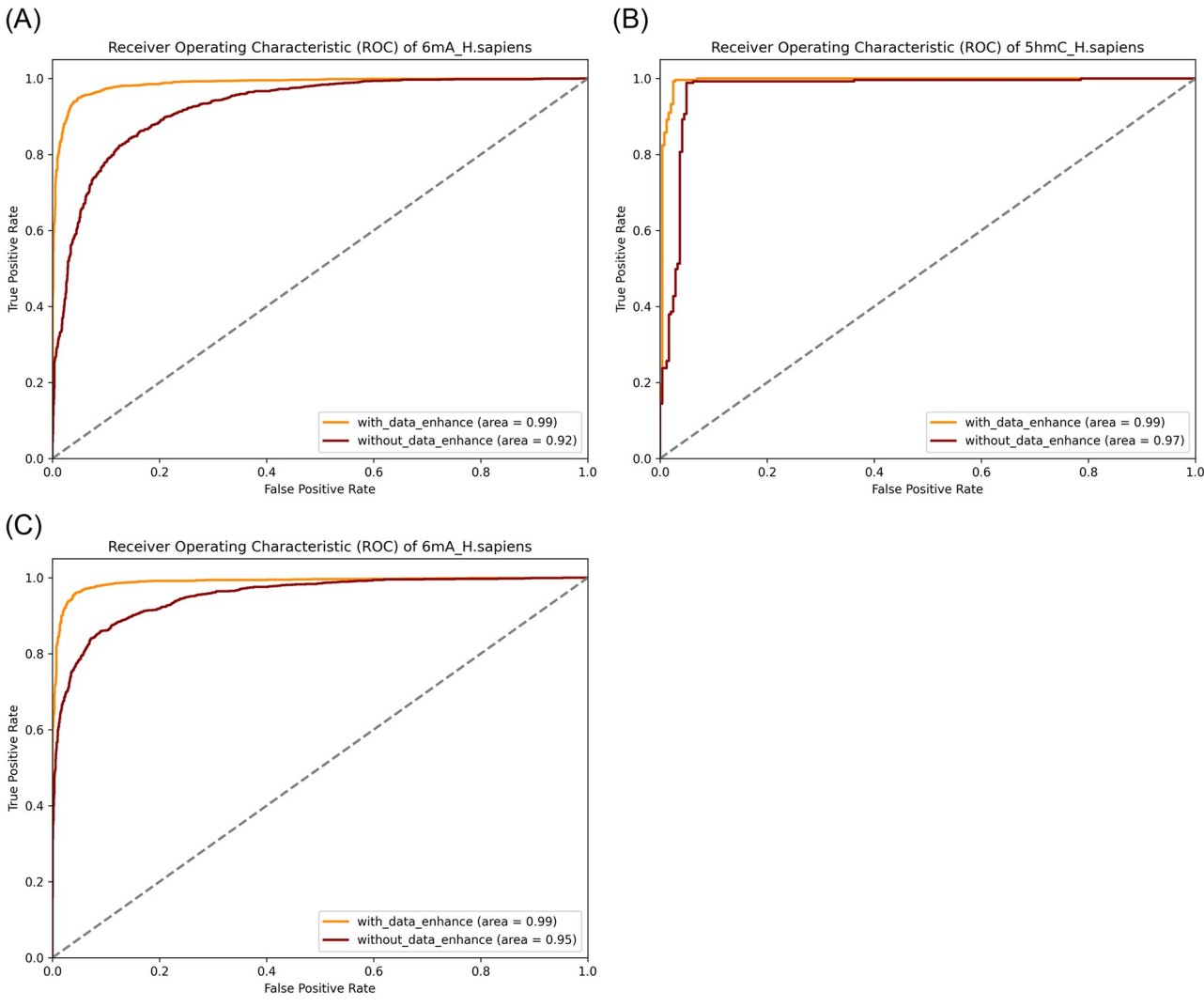

**Fig 4. The ROC curve of with and without data enhance strategy.**

initially intertwined, as shown in Fig 5A and 5C. Still, when the training iterations (epoch) increased to 100, they were separated into distinct categories clearly (as depicted in Fig 5B and 5D). This demonstrates the model's capability to differentiate between DNA methylation and non-DNA methylation samples effectively.

## 4.4 iDNA-ITLM surpasses the current outstanding models

In order to assess the effectiveness of iDNA-ITLM, we conduct a comparative analysis by pitting it against four outstanding predictors, which include iDNA-ABT [37], iDNA-ABF [38], iDNA-MS [39], and MM-6mAPred [40]. Among the four predictors, namely iDNA-ABT, iDNA-ABF, and iDNA-MS, are versatile predictors designed for various methylation prediction tasks. while the MM-6mAPred was initially developed specifically for predicting 6mA site. The rationale behind this lies in the model's adaptability and its ability to be effectively expanded to handle various methylation predictions, including 5hmC and 4mC, in addition to 6mA. Every predictor being compared is trained separately on seventeen training datasets that

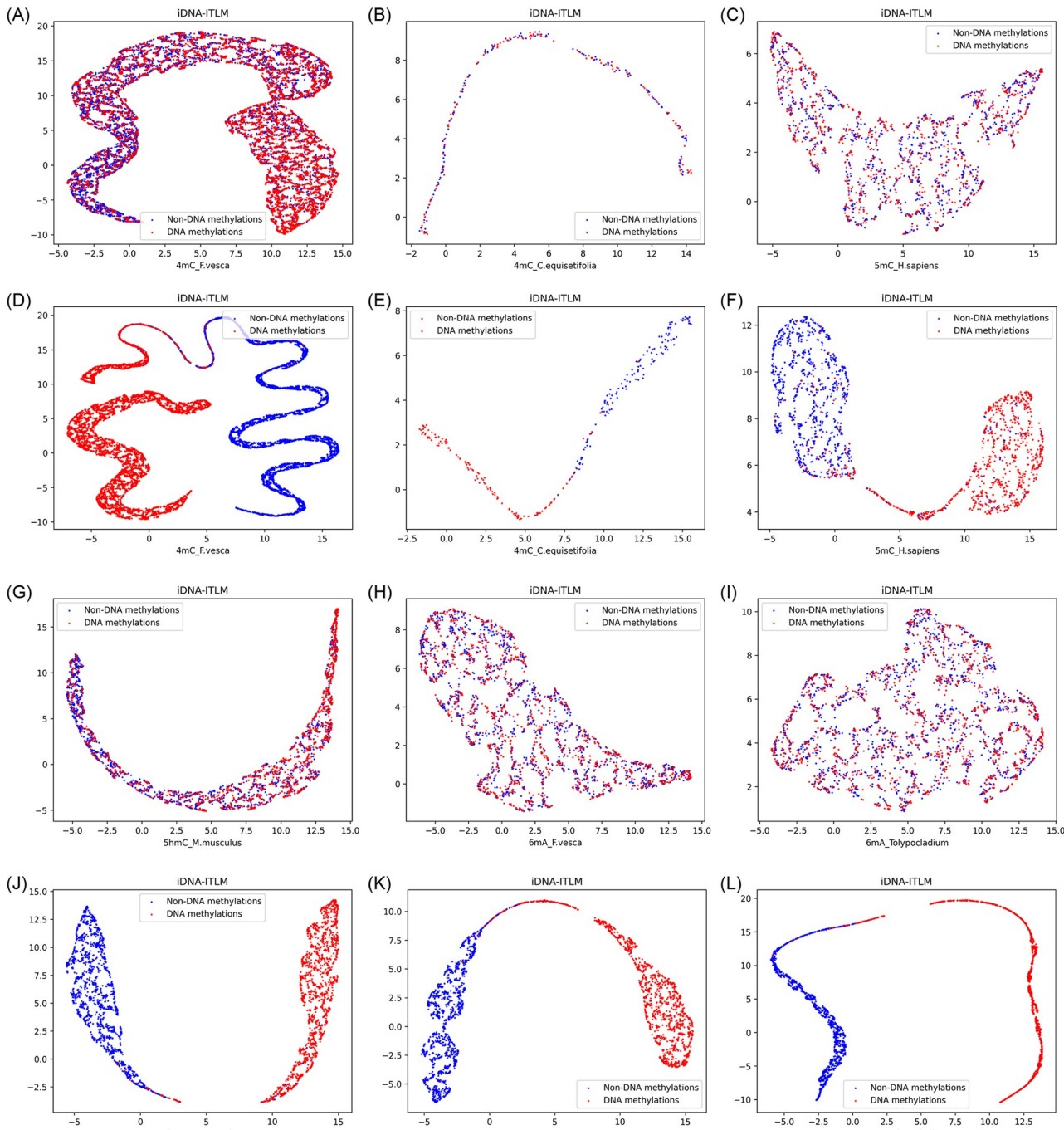

**Fig 5. The visualized distribution of positive and negative samples using UMAP of our model iDNA-ITLM.** In the figures, blue corresponds to non-DNA methylation (negatives), while red corresponds to DNA-methylation (positives). A and C displays the outcomes of the visualizations at epoch 1, while B and D exhibit the outcomes of the visualizations at epoch 100 in 4mC_F.vesca, 4mC_equisetifolia, 5hmC_H.sapiens, 5mC_M.musculus, 6mA_F. vesca and 6mA_Tolypocladium, separately.

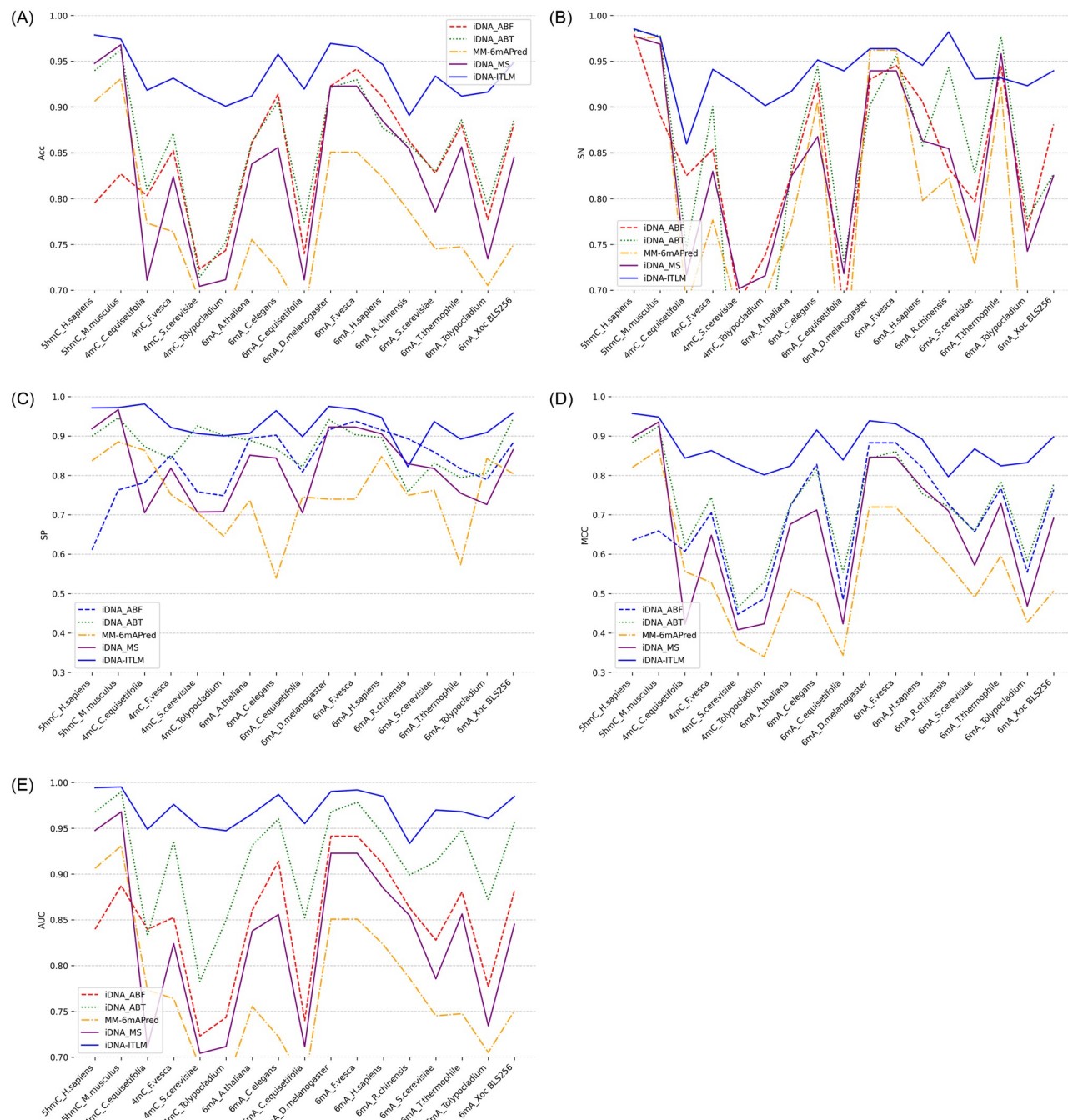

**Fig 6. Performance assessment comparing iDNA-ITLM with other outstanding methods.** A to E correspond to the ACC, SN, SP, MCC, and AUC metrics, indicating the values for our proposed module and other modules, including iDNA-ITLM, iDNA-ABT, iDNA-ABF, iDNA-MS, and MM-6mAPred, across a set of 17 distinct benchmark datasets, individually.

cover various species and types of methylation. Evaluation is then conducted on the respective testing datasets (refer to Table 1 for further details). The evaluation outcomes, including ACC, SN, SP, AUC and MCC are illustrated in Fig 6A to 6E, respectively. As seen in Fig 6B to 6C, in addition to the SN indicator of the model in 6mA_T.thermophile dataset and the SP indicator

of the model in 6mA_R.chinensis dataset, the indicators in all species of our model demonstrates superior performance compared to the four existing predictors across all 17 datasets. This can be attributed to the perfect combination of data enhancement strategy and embedding to the high-dimensional matrix, allowing the model to extract rich features for precise site discrimination.

## 4.5 The cross-species validation has good results

With the aim of investigate the potential applicability of our proposed model between different species, it becomes crucial to establish the capability of a model, train the model using data from one species, and apply it to identify modification sites in other species. Considering this objective, we train separate models tailored to species-specific, utilizing their corresponding 5hmC, 6mA, or 4mC data. Subsequently, we evaluate the performance of these models by testing them with 4mC, 5hmC, or 6mA data from various other species. The cross-validation outcomes are depicted in Fig 7.

As depicted in Fig 7A, when considering the data from H. sapiens and M. musculus as distinct test sets, we assess models based on H. sapiens and M. musculus, yielding satisfactory outcomes. The obtained accuracy rates for H. sapiens and M. musculus reach 97.85% and 97.40%, separately, which indicates the effectiveness of our proposed approach. As illustrated in Fig 7B, the highest accuracy is consistently achieved by the model constructed using its data. However, the outcome derived from the F. vesca model falls slightly short of expectations, exhibiting an accuracy of 87.12%. Additionally, as illustrated in Fig 7C, we can find the ACC of each species model is quite high.

The results show that our deep model performs exceptionally well in dataset cross-validation, with their performance remaining stable across different species. This indicates their ability to achieve excellent results under varying data distributions. This reflects the reliability and robustness of deep learning in various application domains (4mC/5hmC/6mA), making it a powerful tool for tackling complex problems. With cross-validation, we can confidently apply our deep model, ensuring their outstanding performance in different DNA Methylation site recognition.

## 4.6 Our model exhibits strong transfer learning capabilities on RNA sequences

The currently published academic papers have studied the models for predicting DNA methylation site, and models for predicting RNA methylation site, separately. The reason for this separation is that when DNA methylation sequence data is used to train RNA methylation site prediction models, the results are unsatisfactory. Similarly, when RNA methylation sequence data is used to train DNA methylation site prediction models, the results are also unsatisfactory. To assess the effectiveness of the proposed DNA model, we incorporate RNA methylation data into our DNA methylation site prediction model for training without altering the model's parameters. The RNA methylation site prediction results demonstrate the model's strong transfer learning capabilities.

In the experimental part of transfer learning, we use Dao datasets (see Table 2) [41], the datasets include a total of 11 datasets, including human brain (h_b), human kidney (h_k), human liver (h_l), and so on. Initially, we pre-train a model on the RNA sequences, followed by fine-tuning it, resulting in another model referred to as the "transfer learning model". The RNA sequence datasets used for transfer learning are as follows:

We train the transfer learning model by incorporating RNA methylation data into the iDNA-ITLM model and record the recognition results of methylation data sites for 11 species

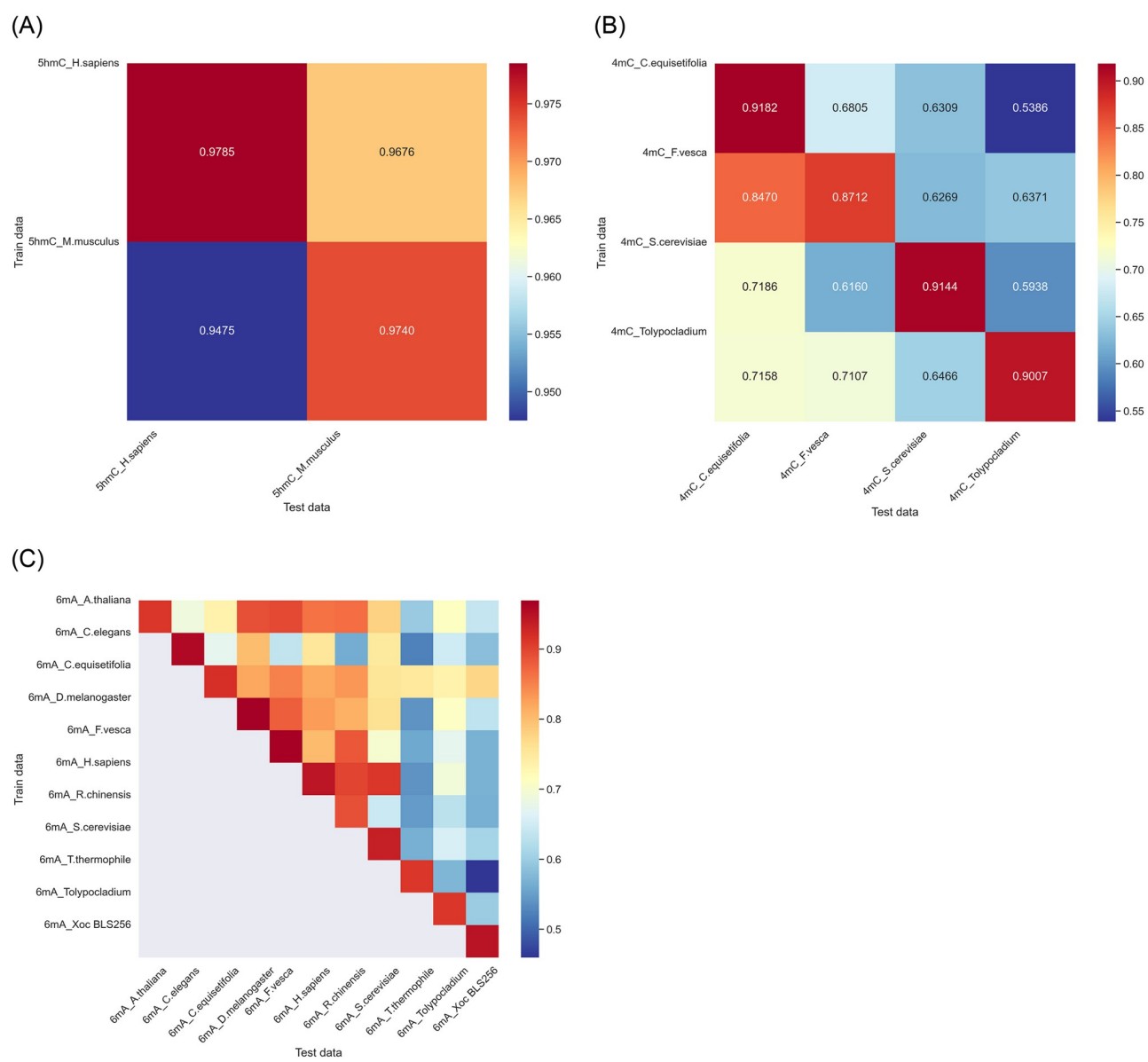

**Fig 7. The heat map of cross-validation.**

in Table 3. It can be seen that the ACC, SN, SP, AUC and MCC evaluation indicators are relatively high, which is better than many RNA methylation sites models, such as M6A-BERT--Stacking [33].

To effectively demonstrate the performance of the proposed approach, we plot ROC curves, Here, the vertical axis corresponds to the true positive rate, while the horizontal axis corresponds to the false positive rate. The results are shown in Fig 8, with the ROC values higher than 0.92 on the independent datasets of 11 species.

In addition, we also compare the method of this paper with other models on the RNA methylation dataset, such as iDNA-LTLM (no fine-tuning), m6A-BERT BiLSTM [33], m6A-BERT ResNet-50 [33]. We randomly selected three species (h_l, m_l, r_l) for model

**Table 2. The datasets of RNA sequence.**

| Species | Tissues | Name | Training | | Independent Testing | |
|---|---|---|---|---|---|---|
| | | | Positive | Negative | Positive | Negative |
| Human | Brain | H_b | 4605 | 4605 | 4604 | 4604 |
| | Kidney | H_k | 4574 | 4574 | 4573 | 4573 |
| | Liver | H_l | 2634 | 2634 | 2634 | 2634 |
| Mouse | Brain | M_b | 8025 | 8025 | 8025 | 8025 |
| | Heart | M_h | 2201 | 2201 | 2200 | 2200 |
| | Kidney | M_K | 3953 | 3953 | 3952 | 3952 |
| | Liver | M_L | 4133 | 4133 | 4133 | 4133 |
| | Testis | M_t | 4707 | 4707 | 4706 | 4706 |
| Rat | Brain | R_b | 2352 | 2352 | 2351 | 2351 |
| | Kidney | R_k | 3433 | 3433 | 3432 | 3432 |
| | Liver | R_l | 1762 | 1762 | 1762 | 1762 |

training. Fig 9 shows the ROC curves of the three species under different models. The area under the ROC curve is a measure of the model's ability to distinguish between the classes. The closer the curve follows the left-hand border and then the top border of the ROC space, the more accurate the test.

From Fig 9, it can be observed that the iDNA-ITLM model demonstrates better ROC curve performance on a randomly selected RNA specie datasets (h_l, m_l, r_l) compared to several other models such as iDNA-LTLM (no fine-tuning), m6A-BERT BiLSTM [33], and m6A-BERT ResNet-50 [33]. The results indicate that the iDNA-ITLM method proposed in this paper also performs well when applied to RNA methylation dataset experiments through transfer learning.

## 5. Conclusion

We propose an iDNA-ITLM model using a novel data enhancement method, combining Embedding, Bert Encoding, Feature extraction, and Classification for identifying DNA methylation sites. We conduct an extensive analysis of our model's predictive performance to assess its reliability and consistency. The experimental outcomes, spanning 17 benchmark datasets

**Table 3. The performances of iDNA-ITLM on RNA methylation datasets.**

| Dataset | ACC | SN | SP | AUC | MCC |
|---|---|---|---|---|---|
| h_b | 0.9036 | 0.9002 | 0.9069 | 0.9489 | 0.8071 |
| h_k | 0.8804 | 0.9099 | 0.8483 | 0.9436 | 0.7609 |
| h_l | 0.9191 | 0.9103 | 0.9277 | 0.9602 | 0.8382 |
| m_b | 0.8729 | 0.9197 | 0.8265 | 0.9401 | 0.7478 |
| m_h | 0.8538 | 0.8958 | 0.8053 | 0.9194 | 0.7063 |
| m_k | 0.8875 | 0.9184 | 0.8874 | 0.9426 | 0.777 |
| m_l | 0.8233 | 0.8347 | 0.8115 | 0.8963 | 0.6465 |
| m_t | 0.8771 | 0.8681 | 0.8861 | 0.9337 | 0.7543 |
| r_b | 0.8677 | 0.9093 | 0.8253 | 0.9261 | 0.7376 |
| r_k | 0.8949 | 0.8683 | 0.9194 | 0.9561 | 0.7898 |
| r_l | 0.9096 | 0.8963 | 0.9211 | 0.9575 | 0.8181 |

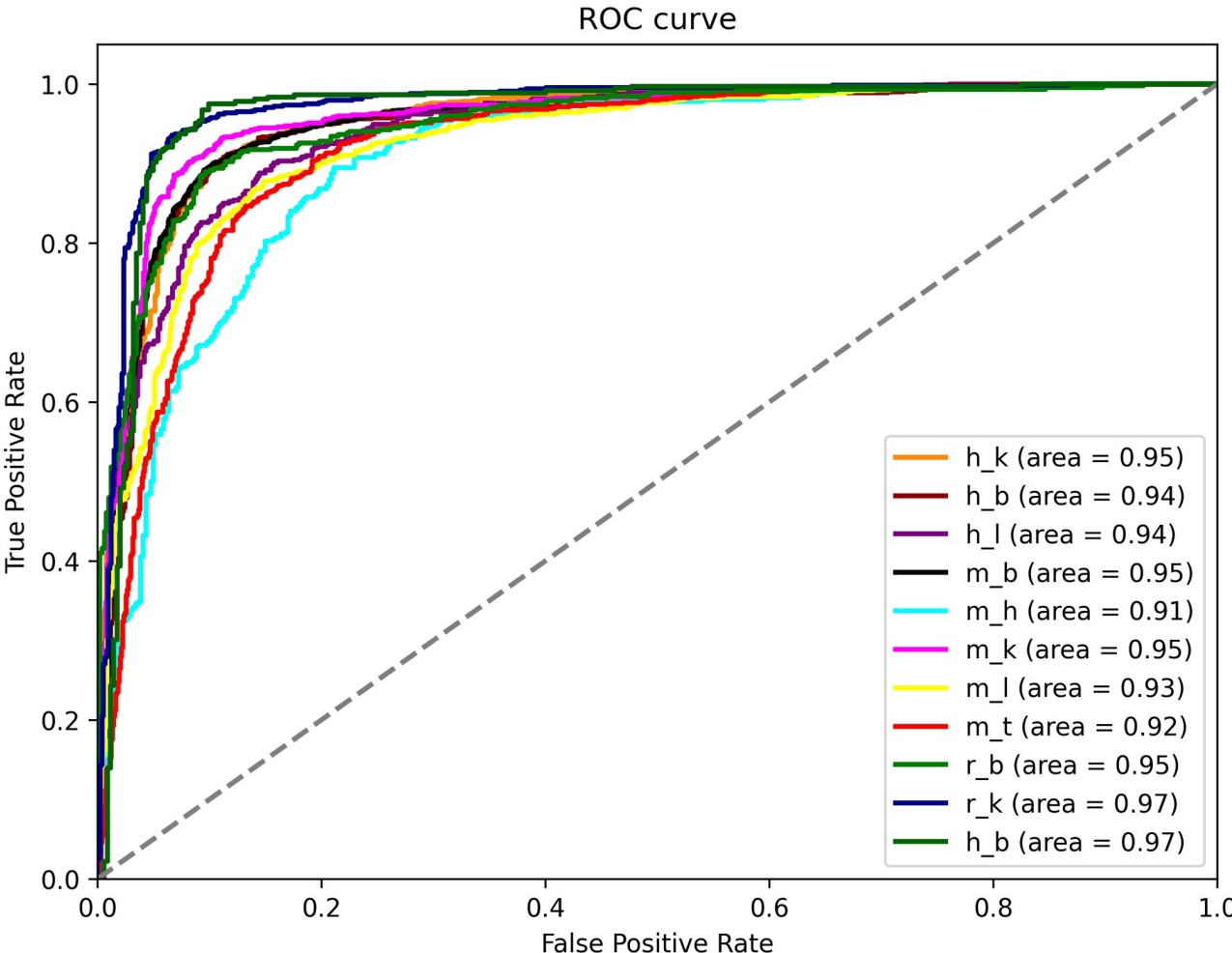

**Fig 8. The ROC curves for detection of RNA methylation sites across multiple tissues in 11 different species.**

encompassing various species and three DNA methylation modifications (4mC, 5hmC, and 6mA), consistently demonstrate that our model outperforms existing sequence-based approaches in terms of both performance and robustness.

Our iDNA-ITLM model enables reasonably precise methylation prediction across a range of species and diverse DNA modifications, including 4mC, 5hmC, and 6mA, and develops an interpretable attention mechanism to delve into the intricacies of DNA methylation mechanisms by data enhancement. This can be understood as continuously self-replicating a short text sequence into a longer one and then embedding it into a large-scale matrix. From the perspective of image processing, transforming a small-scale text into a large-scale matrix enlarges the receptive field, allowing the model to learn more features.

In addition, our model can transfer learning to RNA methylation sequences and produce good results without modifying the hyperparameters in the model. Unlike other models, such as iDNA-ABF, our model is a universal predictor, employing identical hyperparameters across all DNA and RNA methylation species.

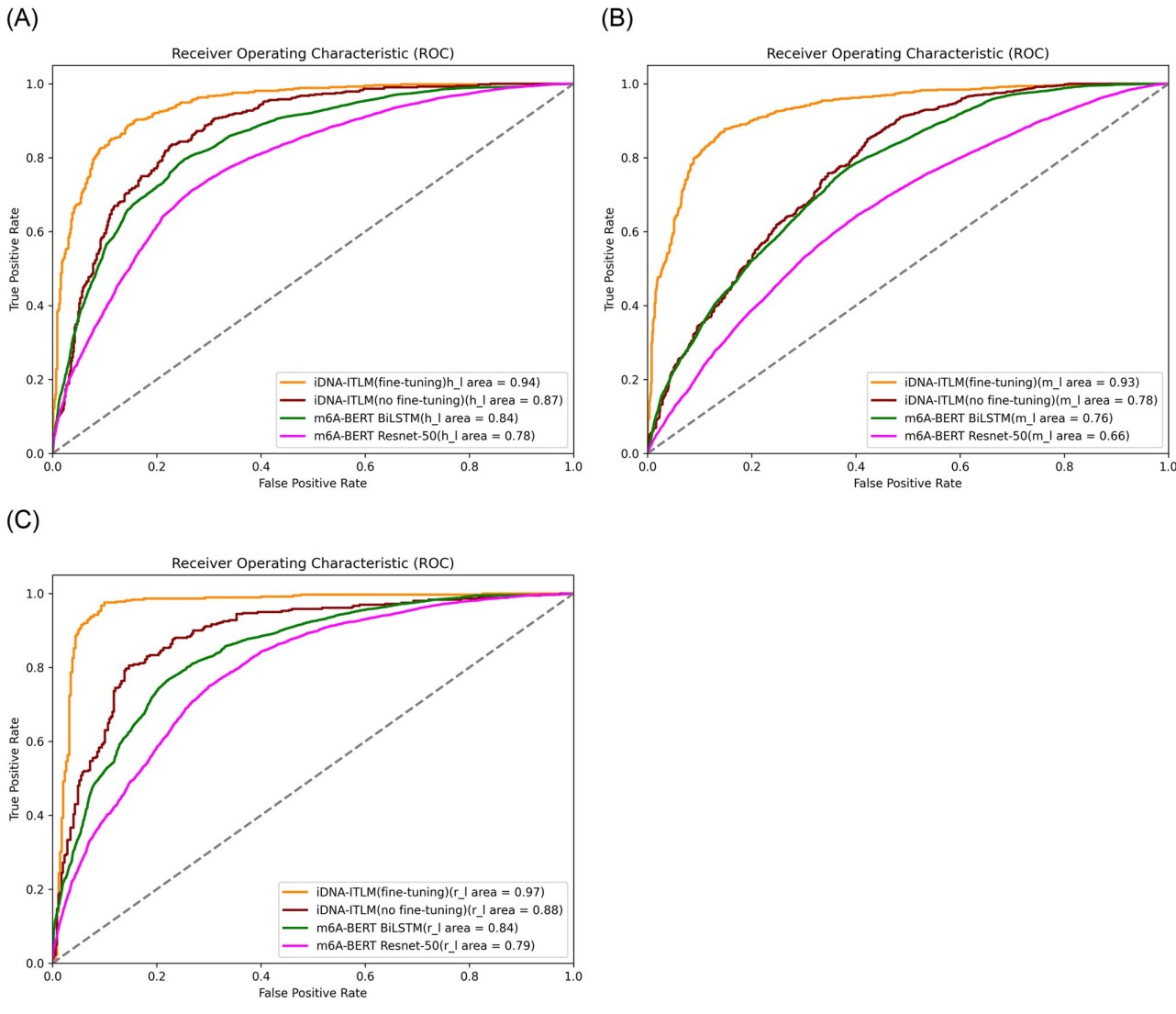

**Fig 9. The ROC of the iDNA-ITLM model with other models on the RNA datasets.**

## Supporting information

**S1 File.**
(RAR)

## Author Contributions

**Conceptualization:** Haixia Long.

**Data curation:** Haixia Long, Rao Zeng.

**Formal analysis:** Zhichao Wang, Haixia Long, Xiling Liu, Bilal Anas.

**Funding acquisition:** Jia Ren.

**Investigation:** Zhichao Wang, Xiling Liu.

**Methodology:** Cui Yani, Haixia Long, Rao Zeng, Jia Ren.

**Resources:** Cui Yani, Zhichao Wang, Haixia Long, Jia Ren.

**Software:** Xia Yu.

**Supervision:** Cui Yani, Xiling Liu, Jia Ren.

**Validation:** Xia Yu.

**Visualization:** Xia Yu.

**Writing – original draft:** Xia Yu.

**Writing – review & editing:** Xia Yu, Haixia Long, Bilal Anas.

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
