## [Decision Letter · Decision Letter 0]

5 Dec 2023

PONE-D-23-33766iDNA-ITLM: An interpretable and transferable learning model for identifying DNA methylationPLOS ONE

Dear Dr. Ren,

Thank you for submitting your manuscript to PLOS ONE. After careful consideration, we feel that it has merit but does not fully meet PLOS ONE’s publication criteria as it currently stands. Therefore, we invite you to submit a revised version of the manuscript that addresses the points raised during the review process.

Please submit your revised manuscript by Jan 19 2024 11:59PM. If you will need more time than this to complete your revisions, please reply to this message or contact the journal office at plosone@plos.org. Please include the following items when submitting your revised manuscript:A rebuttal letter that responds to each point raised by the academic editor and reviewer(s). You should upload this letter as a separate file labeled 'Response to Reviewers'.A marked-up copy of your manuscript that highlights changes made to the original version. You should upload this as a separate file labeled 'Revised Manuscript with Track Changes'.An unmarked version of your revised paper without tracked changes. You should upload this as a separate file labeled 'Manuscript'.

We look forward to receiving your revised manuscript.

Kind regards,

Li Chen

Academic Editor

PLOS ONE

Reviewers' comments:

Reviewer's Responses to Questions

**Comments to the Author**

1. Is the manuscript technically sound, and do the data support the conclusions?

Reviewer #1: Partly

Reviewer #2: Partly

2. Has the statistical analysis been performed appropriately and rigorously? 

Reviewer #1: Yes

Reviewer #2: N/A

3. Have the authors made all data underlying the findings in their manuscript fully available?

Reviewer #1: No

Reviewer #2: Yes

4. Is the manuscript presented in an intelligible fashion and written in standard English?

Reviewer #1: Yes

Reviewer #2: No

5. Review Comments to the Author

Reviewer #1: The manuscript by Yu et al. reports a novel data augmentation strategy, which involves the continuous self-replication of a short DNA sequence into a longer one, followed by its embedding into a high-dimensional matrix to expand the receptive field. This approach, combined with the existing BERT model, is used for predicting DNA methylation sites. The model performs well across 17 datasets from different species, and it can also be extended to predict RNA methylation.

Generally, the manuscript is well written, and the conclusions are supported by the presented results. However, some issues that need to be addressed to provide more substantial evidence of the model's superiority and to elucidate the entire training process.

Major:

1. The selling point of this paper is the data enhancement part. Did you compare the model performance with and without data enhance strategy? Adding this experiment will make the results more convincing.

2. How did you train your model? As you used pretrained BERT model, did you retrain it using your data or freeze the parameters of it? Please provide more information to clarify your training process.

3. For transfer learning part, did you fine-tune the pre-trained DNA model using RNA sequences or train it from scratch? Did you retrain the whole model or only retrain a portion of the model? Please clarify your transfer learning process.

Minor:

1. Check for typos throughout the text. Eg: the title of MLM section

2. Please provide higher quality figures. When zoomed in, the figures lose clarity and appear blurry.

3. The formatting of the formulas and equations is inconsistent (Italic? Bold?) and the layout is messy. Please modify.

4. For Figure6, I guess the xlabel and ylabel are ‘train data’ and ‘test data’? Please add labels of axis.

5. In transfer learning part, you claimed you did transfer learning without altering model’s parameters. I guess you're referring to hyperparameters here?

Reviewer #2: This manuscript by Xia et al introduces a novel data enhancement strategy for identifying DNA methylation sites. The authors demonstrate that the proposed method outperforms baseline methods, including iDNA-ABT, iDNA-ABF, iDNA-MS, and MM-6mAPred. While the paper is well-organized, there are concerns about the clarity in describing methods and results.

Comments:

1. The paper contains numerous grammar errors, confusing sentences/subtitles, and misreferences (e.g., Figure1B referred to as Fig2B). A careful review of the paper for these issues is necessary before submission.

2. The motivation behind the self-replicating operation of the sequence is unclear. Does the model benefit from this operation, or is it solely for converting the sequence to a matrix? The authors should clarify this aspect.

3. Figure 1 is confusing. Does Fig 1.B involve data enhancement through self-replicating? If so, how is a sequence converted to a matrix with different rows? Subfigures CDEF can be consolidated into a single figure. Additionally, the species information in the prediction in Figure H needs clarification—are these species details part of the training data or the model? The authors need to re-draw this overview figure to make it clear.

4. How was the training / testing split performed? Why are they equal?

5. The motivation for presenting the logo plot of methylation in Figure 2 is not clear. Authors should provide more context. If not, I would suggest to remove the figure2

6. For Figure 5, if the datasets are uncorrelated, a bar plot is suggested instead of a line plot.

7. In the transfer learning section, authors should include the performance of baseline methods (directly trained on RNA dataset without fine-tuning)

6. PLOS authors have the option to publish the peer review history of their article (what does this mean?). If published, this will include your full peer review and any attached files.

Reviewer #1: No

Reviewer #2: No

---

## [Author Response · Author response to Decision Letter 0]

29 Dec 2023

Dear Prof. reviewer，

 Thank you very much for your comments and professional advice. These help us to improve academic rigor of our article. Based on your suggestions and requests, we made corrected modifications on the revised manuscript. Furthermore, we would like to show the details as follows:

Reviewer 1

The manuscript by Yu et al. reports a novel data augmentation strategy, which involves the continuous self-replication of a short DNA sequence into a longer one, followed by its embedding into a high-dimensional matrix to expand the receptive field. This approach, combined with the existing BERT model, is used for predicting DNA methylation sites. The model performs well across 17 datasets from different species, and it can also be extended to predict RNA methylation.

Generally, the manuscript is well written, and the conclusions are supported by the presented results. However, some issues that need to be addressed to provide more substantial evidence of the model's superiority and to elucidate the entire training process.

Major:

1. The selling point of this paper is the data enhancement part. Did you compare the model performance with and without data enhance strategy? Adding this experiment will make the results more convincing.

The author’s answer: We have supplemented the experiment comparing the performance of the model with and without data enhancement in Section 4.2 of the article. (Page 15)

2. How did you train your model? As you used pretrained BERT model, did you retrain it using your data or freeze the parameters of it? Please provide more information to clarify your training process.

The author’s answer: When using the Bert model, we retrained it with on one or several species DNA methylation sequence data. During the process of retraining the model, we fine-tuned relevant parameters, such as the learning rate and embedding, to optimize the model's performance. Once the model's parameters were well-trained, this set of parameters could be applied to the identification of methylation site in other DNA and RNA species, with equally good performance.

3. For transfer learning part, did you fine-tune the pre-trained DNA model using RNA sequences or train it from scratch? Did you retrain the whole model or only retrain a portion of the model? Please clarify your transfer learning process.

The author’s answer: In the transfer learning section, we did not fine-tune the pre-trained DNA model using RNA sequences. Instead, we input RNA methylation sequences into the model proposed in this paper, employing the fine-tuning parameters that optimized the model during the DNA methylation site recognition stage. This means that methylation sequences from different RNA species are directly trained in the model without the need for fine-tuning parameters for different species, achieving good performance.

Minor:

1. Check for typos throughout the text. Eg: the title of MLM section

The author’s answer: Thank you for pointing out the issue, we have now modified it to a consistent format. We have made corrections to the typos.（Page 7）

2. Please provide higher quality figures. When zoomed in, the figures lose clarity and appear blurry.

The author’s answer: Thank you for pointing out the issue, all the images in the article are output in a 300dpi format. I assume the distortion you're referring to upon enlargement is regarding Figure 2. This figure is a visual representation converted from sequence data after data augmentation into matrix data. Unlike other images, it does not possess distinctive features, so visually, it appears distorted when enlarged.

3. The formatting of the formulas and equations is inconsistent (Italic? Bold?) and the layout is messy. Please modify.

The author’s answer: Thank you for pointing out the issue, We have already standardized the formatting of the equations in the article, uniformly setting them to Cambria Math 12.（Page 9-10）

4. For Figure6, I guess the xlabel and ylabel are ‘train data’ and ‘test data’? Please add labels of axis.

The author’s answer: Yes, the xlabel and ylabel are ‘train data’ and ‘test data’ for Figure 6. We have already added the corresponding xlabel and ylabel. (Page 18)

5. In transfer learning part, you claimed you did transfer learning without altering model’s parameters. I guess you're referring to hyperparameters here?

The author’s answer: Yes, the model's parameters referred to here are the hyperparameters.

Reviewer 2

This manuscript by Xia et al introduces a novel data enhancement strategy for identifying DNA methylation sites. The authors demonstrate that the proposed method outperforms baseline methods, including iDNA-ABT, iDNA-ABF, iDNA-MS, and MM-6mAPred. While the paper is well-organized, there are concerns about the clarity in describing methods and results.

Comments:

1. The paper contains numerous grammar errors, confusing sentences/subtitles, and misreferences (e.g., Figure1B referred to as Fig2B). A careful review of the paper for these issues is necessary before submission. 

The author’s answer: Thank you for pointing out the issue, We have already corrected the relevant errors in the article. (Page 4)

2. The motivation behind the self-replicating operation of the sequence is unclear. Does the model benefit from this operation, or is it solely for converting the sequence to a matrix? The authors should clarify this aspect.

The author’s answer: The motivation for the self-replication of the sequence is that the length of the DNA methylation sequence is relatively short, only 41bp, this led the author to think of the receptive field in image feature extraction, where the larger the receptive field, the richer the features extracted. Therefore, the DNA methylation sequence is self-replicated and embedded into a large-scale matrix, thereby increasing the sequence's receptive field to facilitate the extraction of features from the DNA methylation sequence. This motivation is explained in the Introduction section of the article. The model benefits from the self-replication of DNA, and the author has supplemented the article with experiments before and after data augmentation. (Page 3, 14-15)

3. Figure 1 is confusing. Does Fig 1.B involve data enhancement through self-replicating? If so, how is a sequence converted to a matrix with different rows? Subfigures CDEF can be consolidated into a single figure. Additionally, the species information in the prediction in Figure H needs clarification—are these species details part of the training data or the model? The authors need to re-draw this overview figure to make it clear.

The author’s answer: ①Yes, Figure 1.B includes data self-replication to achieve data augmentation, and then through the embedding module, it realizes the conversion of data from sequences to matrices. Each row of sequence data is transformed into a matrix after going through self-replication and the embedding module. The relevant detailed information is thoroughly explained in the 2.3 embedding module section. (Page 5)

②The species information in Figure 1.H indicates that the species category can be directly determined after training the model with the training data, which consists of DNA methylation sequence data from different species.

③ To make the Figure 1 look clearer, we have made slight adjustments to Figure 1. (Page 4)

4. How was the training / testing split performed? Why are they equal?

The author’s answer: The training and test sets have an equal number of samples because this is how it is set up in the iDNA-MS database. In this database, for all species, the number of samples in the training set and the test set is the same. This facilitates cross-validation, making the experimental design simpler and more consistent. Some other papers, such as iDNA-ABF, iDNA-MS, also have their training and test sets set up in this manner.

5. The motivation for presenting the logo plot of methylation in Figure 2 is not clear. Authors should provide more context. If not, I would suggest to remove the figure2

The author’s answer: We utilized the WebLogo tool to illustrate the DNA methylation sequence patterns of different species, as shown in Figure 2. From the figure, it is noticeable that there are significant differences in the positions and extents of base enrichment for various methylation types within the genome. This provides a theoretical basis for further research into models for DNA methylation site recognition. It demonstrates that sequences are not random or devoid of distinct features; rather, DNA sequences from different species exhibit unique characteristics. This concept is also elaborated upon in the article. (Page 11)

6. For Figure 5, if the datasets are uncorrelated, a bar plot is suggested instead of a line plot.

The author’s answer: The dataset used by the model in the Figure 5 is relevant and all from the same dataset.Therefore, we used a line plot.(Page 16)

7. In the transfer learning section, authors should include the performance of baseline methods (directly trained on RNA dataset without fine-tuning)

The author’s answer: Thank you for pointing out the issue, we have already replenished the experiments. (Page 21-22)

---

## [Decision Letter · Decision Letter 1]

24 Mar 2024

iDNA-ITLM: An interpretable and transferable learning model for identifying DNA methylation

PONE-D-23-33766R1

Dear Dr. Ren,

We’re pleased to inform you that your manuscript has been judged scientifically suitable for publication and will be formally accepted for publication once it meets all outstanding technical requirements.

Kind regards,

Li Chen

Academic Editor

PLOS ONE

Additional Editor Comments (optional):

Reviewers' comments:

Reviewer's Responses to Questions

**Comments to the Author**

1. If the authors have adequately addressed your comments raised in a previous round of review and you feel that this manuscript is now acceptable for publication, you may indicate that here to bypass the “Comments to the Author” section, enter your conflict of interest statement in the “Confidential to Editor” section, and submit your "Accept" recommendation.

Reviewer #1: All comments have been addressed

Reviewer #2: All comments have been addressed

2. Is the manuscript technically sound, and do the data support the conclusions?

Reviewer #1: (No Response)

Reviewer #2: Yes

3. Has the statistical analysis been performed appropriately and rigorously? 

Reviewer #1: (No Response)

Reviewer #2: Yes

4. Have the authors made all data underlying the findings in their manuscript fully available?

Reviewer #1: (No Response)

Reviewer #2: Yes

5. Is the manuscript presented in an intelligible fashion and written in standard English?

Reviewer #1: (No Response)

Reviewer #2: Yes

6. Review Comments to the Author

Reviewer #1: (No Response)

Reviewer #2: (No Response)

7. PLOS authors have the option to publish the peer review history of their article (what does this mean?). If published, this will include your full peer review and any attached files.

Reviewer #1: No

Reviewer #2: No

---

## [Editor Report · Acceptance letter]

29 Apr 2024

PONE-D-23-33766R1 

PLOS ONE

Dear Dr. Ren, 

I'm pleased to inform you that your manuscript has been deemed suitable for publication in PLOS ONE. Congratulations! Your manuscript is now being handed over to our production team.

Kind regards, 

on behalf of

Dr. Li Chen 

Academic Editor

PLOS ONE